# Pain Reduction with Repeated Injections of Botulinum Toxin A in Upper Limb Spasticity: A Longitudinal Analysis from the ULIS-III Study

**DOI:** 10.3390/toxins17030117

**Published:** 2025-03-01

**Authors:** Lynne Turner-Stokes, Khan Buchwald, Stephen A. Ashford, Klemens Fheodoroff, Jorge Jacinto, Ajit Narayanan, Richard J. Siegert

**Affiliations:** 1Department of Palliative Care, Policy and Rehabilitation, Cicely Saunders Institute, Florence Nightingale Faculty of Nursing, Midwifery and Palliative Care, King’s College London, London SE5 9PJ, UK; stephen.ashford@nhs.net; 2Regional Hyper-Acute Rehabilitation Unit, Northwick Park Hospital, London North West University Healthcare NHS Trust, London HA1 3UJ, UK; 3School of Clinical Sciences, Faculty of Health and Environmental Science, Auckland University of Technology, Auckland 0627, New Zealand; khan.buchwald-mackintosh@aut.ac.nz; 4Neurorehabilitation, Gailtal-Klinik, 9620 Hermagor, Austria; klemens.fheodoroff@me.com; 5Centro de Medicina de Reabilitaçãode Alcoitão, Serviço de Reabilitação de Adultos 3, 2649-506 Alcabideche, Portugal; 6School of Engineering, Computer and Mathematical Sciences, Auckland University of Technology, Auckland 1010, New Zealand; ajit.narayanan@aut.ac.nz; 7Department of Psychology and Neuroscience, Faculty of Health and Environmental Science, Auckland University of Technology, Auckland 0627, New Zealand; richard.siegert@aut.ac.nz

**Keywords:** pain, botulinum toxin, upper limb spasticity, goal attainment scaling, rehabilitation

## Abstract

Pain reduction is a common goal of the treatment of upper limb spasticity with botulinum toxin (BoNT-A). ULIS-III was a large international, observational, longitudinal study (N = 953) conducted in real-life clinical practice over two years. In this secondary post hoc analysis, we examine whether goals for pain reduction were met over repeated injection cycles. We report serial changes in pain severity and explore predictors of pain reduction and injection frequency. Patients were selected if pain reduction was a primary/secondary goal for at least one cycle (n = 438/953). They were assessed at the start and end of each cycle using the goal attainment T-score alongside a self-report of pain severity (range 0–10). Across all cycles, pain-related goals were set for 1189/1838 injections (64.7%) and were achieved in 839 (70.6%). Patients continued to show a significant reduction in pain (*p* < 0.001) for each injection up to seven cycles, with some cumulative benefit (*p* < 0.001). Those requiring more frequent injections tended to have higher starting pain scores and a smaller reduction in pain score, but these differences were not significant when other covariates (age, previous injection history, time since onset, severity and distribution of spasticity) were taken into account (*p* > 0.713). Conclusion: Repeated BoNT-A administration continued to result in a significant reduction in upper limb spasticity-related pain, regardless of patient-related factors.

## 1. Introduction

Spasticity is a common consequence of damage to the central nervous system, which is characterised by muscle overactivity leading to stiffness, pain, contractures and deformity [1]. Injection with botulinum toxin type A (BoNT-A) is shown to be a safe and effective focal intervention to reduce spasticity [2,3,4,5] and is recommended for use in routine clinical practice by national and international guidelines [6,7,8].

Spasticity-related pain may be multifactorial [9,10,11,12]. It may include pain from the muscle overactivity itself, as well as pain due to joints being pulled out of alignment and tension in the soft tissues (which itself can cause hypertonia) [13,14,15,16]. In addition, patients with damage to the central nervous system may have other causes of pain, including neuropathic pain [17,18] and complex regional pain syndrome [19,20]. The relationship between spasticity and pain is compounded by the fact that pain increases spasticity, creating a spiralling course of more pain and disability [12].

Suitably targeted management of spasticity with BoNT-A injection and other concomitant therapies may help to relieve pain by reducing muscle overactivity, restoring joint alignment and reducing tension and deformity [21]. In addition, there are other potential mechanisms by which botulinum toxin could directly or indirectly alter pain, including (1) changes in the sensitivity and response patterns of muscle nociceptors, (2) changes in muscle spindle afferents, (3) alterations in the cholinergic control of vascular and autonomic functions, including neurogenic inflammation, (4) induced neuroplastic changes in the processing of afferent somatosensory activity and (5) direct non-cholinergic effects on pain afferents [22,23,24]. There are few specific studies on the prevalence of pain in upper limb spasticity, but the overall prevalence of pain in spasticity is around 65% [25], and pain reduction is an increasingly recognised goal for the treatment of spasticity [25,26,27,28].

Although some trials and meta-analyses have not demonstrated a significant reduction in pain symptoms following BoNT-A injection [29,30], this does not resonate with clinical experience. Open-label studies [27,28,31,32] focusing specifically on this symptom have shown clinically important improvements in pain from a single cycle of BoNT-A, and a meta-analysis of nine trials showed significant improvements in hemiplegic shoulder pain compared with placebo [33]. A recent pooled-sample analysis has also suggested progressive improvement in pain over repeated cycles [27], which raises a question about how often injection may need to be repeated.

Spasticity is not always painful, however, so one possible reason for failure to demonstrate significant overall pain reduction in some trials is that this may not be a relevant outcome for a significant proportion of patients [8]. Goal attainment scaling (GAS) [34] is increasingly accepted as a person-centred outcome measure for evaluating the attainment of an individual’s priority goals for treatment and is now widely used to assess the effectiveness of rehabilitation in the areas that matter most to the patients and their caregivers [35,36,37,38,39]. It can therefore help to identify and select those patients for whom pain reduction is identified specifically as a goal for treatment.

Conducted over more than a decade, the Upper Limb International Spasticity (ULIS) programme is a series of international observational studies designed to describe and evaluate real-life clinical practice regarding the use of BoNT-A to manage upper limb spasticity [26]. A key feature of the programme is that it incorporates GAS as the primary outcome measure, collected alongside standardised validated measures [26,40]. Studies from within the programme have confirmed the feasibility of a common international dataset to collect systematic prospective data and of using GAS to capture person-centred outcomes relating to specific symptoms, including pain [41]. They have shown that pain management is increasingly recognised as a primary goal for treatment in routine clinical practice [26] and that goals for pain reduction were achieved in over 80% of patients [41,42,43].

Spasticity is a long-term condition, and longitudinal studies have demonstrated the benefits of repeated injections of BoNT-A over time for a variety of symptoms [42,44,45,46,47,48]. ULIS-III [40,42] (the third study in the ULIS series) is one of the largest multi-centre longitudinal studies of the effects of repeated injection of BoNT-A for upper limb spasticity to be conducted to date, with a total ‘effectiveness population’ of N = 953. ULIS-III introduced the Upper Limb Spasticity Index (ULSI), which records GAS alongside a limited set of standardised measures, selected specifically according to the priority goal area(s) for treatment in each cycle. Pain severity is measured by self-report using the Numbered Graphic Rating Scale (NGRS) in a range of 0–10 [40]. The results of the main study provide evidence on a population level for the continued response to repeated injection over a 2-year period with respect to goal attainment, pain and involuntary movement, as well as improved ability to use and care for the affected limb [42]. A post hoc analysis identified potential predictors of positive response to treatment as being the use of injection guidance techniques, female gender and some toxin preparations (abobotulinum over onabotulinum) [49].

The aim of this further post hoc analysis was to explore in greater depth the outcomes from repeated injections of BoNT-A in patients for whom upper limb spasticity-related pain was specifically identified as a goal for treatment in real-life clinical practice. The objectives were as follows:(a)To determine the extent to which goals for pain reduction continued to be met through repeated cycles of injection over the 2-year period;(b)To examine serial changes in reported pain severity in order to identify any cumulative effect on pre- or post-injection pain scores;(c)To explore any patient-related factors that may determine which patients require more or less frequent injections;(d)To examine any patient-related predictors of pain reduction.

On a clinical level, we hypothesised that patient-related factors that predict pain reduction or the need for more frequent injections might be age, gender, the time since injury, previous injection with BoNT-A and the severity and/or distribution of spasticity.

## 2. Results

Of the 953 patients in the ULIS-III’s ‘effectiveness population’, 438 (46%) had at least one pain-related goal during the study and formed the ‘pain subgroup’ for this analysis.

### 2.1. Demographics

The demographics are shown in Table 1. They were not significantly different for the pain subgroup and the patients without any pain-related goals, except that, as expected, the pain subgroup had a higher mean baseline pain score on the Disability Assessment Scale (DAS) (adjusted *p* < 0.001). They also had a higher proportion of regional spasticity (*p* < 0.001) with slightly more severe proximal MAS scores (*p* = 0.014). Within the pain subgroup, the male/female ratio was 52:48%; 391 (89%) had an acquired brain injury (most commonly following stroke (364/391 = 93%)) and the mean time from onset was 7.6 years (SD = 9.6), indicating mainly chronic spasticity. Eighty-six percent had regional spasticity, and two-thirds of the group (66%) had previously received BoNT-A injections for upper limb spasticity.

During the 2-year period, there were a total of 1838 injections given to the pain subgroup, of which 1189 (64.7%) had a specified pain goal. This percentage ranged from 59% to 73% across the eight cycles (see Figure A1 and Table A1 in Appendix B). The percentage of pain-related goals that were achieved ranged from 62 to 75% across the first six cycles but dropped off to 52% and 20% in cycles 7 and 8, respectively.

### 2.2. Cycle-by-Cycle Analysis of GAS T-Scores and Pain Scores

The first step of this evaluation was a cycle-by-cycle analysis (at the population level) of mean goal attainment and pain severity scores for all the patients who had a pain-related goal in any cycle, across the eight cycles. Figure 1a,b show the mean (95% bootstrapped CI) GAS T-scores and NGRS pain scores for all the patients who had a pain-related goal in any cycle before and after injection and the change in scores for each cycle. (The values and paired *t*-tests are given in Table A2 in Appendix B, with *p*-values adjusted for multiple tests using the Holm–Bonferroni method.)

The figures show a significant increase in GAS T-score and reduction in NGRS pain score between the start and end of each cycle (i.e., before and after injection), which remained significant across all eight cycles (adjusted *p* < 0.002 and <0.015, respectively), confirming that patients continued to derive pain relief from each injection, regardless of how many cycles they received.

The magnitude-of-change scores appear to decrease across the eight cycles from 13.3 to 7.9 for the GAS T-scores and −2.8 to −1.8 for the NGRS scores (Table A2). However, this could just be a sampling characteristic, given that the number of patients progressively diminished from cycles one to eight.

### 2.3. Serial Changes in Reported Pain Severity with Number of Cycles

To explore the effects of repeated injection more directly, changes in reported pain severity across cycles were examined by the number of injection cycles (1–8) with a serial pain score recorded. Figure 2 shows the mean NGRS scores for the start and end of each cycle and the change in scores within each cycle, grouped by the number of cycles. The raw means and 95% bootstrapped CIs are available in Appendix A.

The first step in this exploratory analysis was a simple visual inspection of the graphic data to examine trends and to sense-check the subsequent statistical analysis. From the line graphs in Figure 2, it can be seen that patients with more frequent injection cycles for pain (5–8) tended to have higher scores at baseline (i.e., at the start of cycle 1, before injection) than those who had only 1–3 cycles during the 2-year period. Moreover, this difference appeared to be directly in proportion to the number of cycles (i.e., the more cycles they had, the higher the mean baseline NGRS score). However, over successive cycles, there is a visible trend towards a reduction in the mean start-of-cycle scores (i.e., before each injection), regardless of the number of cycles a patient received, suggesting some cumulative effect of repeated injections.

Similarly, the mean end-of-cycle scores (after BoNT-A injection) in cycle 1 were also proportionately higher in relation to cycle frequency. Again, there is a trend towards a reduction in mean end-of-cycle scores with repeated injection, at least across the first three cycles. After three cycles, they appear to plateau.

Generally, there is a trend towards smaller change scores with repeated cycles, at least up to cycle 5. In contrast, patients who had eight cycles appeared to show an anomalous (but quite striking) progressive increase in change scores over the first five cycles. However, this latter finding should be interpreted with caution, given the small number of patients in this group (n = 4, which is <1% of the analysed population group).

To examine effect sizes and whether there was a significant difference in pain scores between cycles, as shown in Figure 2, the NGRS start and end scores, as well as the change scores, for each cycle were further explored using repeated-measures ANOVAs, grouped by the number of cycles each patient had (see Table 2). Although all three ANOVAs reached statistical significance for at least some cycles, the effect sizes were only ‘small to moderate’, except in patients requiring a higher frequency of injection.

### 2.4. Cycle-Frequency Group Comparison

In order to examine the apparent trends in responses between patients requiring more and less frequent injections statistically, patients were then divided into three ‘cycle-frequency’ groups: group 1 (1–3 cycles only, n = 164), group 2 (4–5 cycles, n = 178) and group 3 (6–8 cycles, n = 96). The demographics for these three groups are given in Table A3 in the Appendix. The only significant between-group differences were that the patients in group 2 were younger (46% <51 years compared with 29–30% in the other two groups (χ^2^ = adjusted *p* = 0.026)), and groups 2 and 3 were significantly more likely to have had previous BoNT-A injection (χ^2^ = 33.0 *p* < 0.001). The trend towards a longer time since onset of injury in group 3 did not reach significance.

Again, starting with a simple visual inspection of the data, Figure 3 presents line graphs of the mean NGRS scores by cycle (both before and after BoNT-A injection) for each of the three cycle-frequency groups (the mean and 95% bootstrapped CI values are given in Table A4). The mean NGRS pain scores are highest for group 3 (indicating more severe pain), especially for the start of cycles 1 and 2. This is also reflected in lower rates of overall goal attainment for group 3 (see Table A4).

NGRS pain scores for groups 1 and 2 appear similar at baseline. However, looking at the trendlines, there is a steeper downward trend in pain scores across the first three cycles for group 1, in comparison to groups 2 and 3, for which the slopes of the trendlines appear to decrease at a similar rate. The end-of-cycle scores appear to be consistently lower for group 2 than group 3 across the first five cycles.

Overall, all three groups show a trend towards an average reduction in pain scores which continues until cycle 8 for those who required that number of injections. However, this simple descriptive analysis does not take account of other covariates, and we were interested to explore any other patient-related factors that might determine which patients require more or less frequent injections of BoNT-A to manage their spasticity-related pain.

### 2.5. Predictors of Pain Scores

Table 3a,b, respectively, present a mixed-effects model to explore factors that may predict NGRS pain scores at both the start and end of a cycle (before and after injection) after taking patient-related factors including age, gender, time since injury, previous injection with BoNT-A and severity/distribution of spasticity into account. These were added as covariates to control for their effect, along with group and cycle number. We also added an interaction term for cycle number and group to examine different slope values, i.e., the rate of change in NGRS score across successive cycles.

In this model, the individual was treated as a random effect to account for dependency between observations in the data. The intercepts were the baseline NGRS scores at (a) the start and (b) the end of the first cycle, taking into account all the covariates included in the model. The average NGRS pain scores at the start and end of the first cycle were 6.45 and 3.4 out of 10 for group 1, amounting to an average reduction of just over 3 points after the first injection of BoNT-A, after taking these patient-related factors into account.

Within the mixed-effects models, after controlling for the patient-related factors, group 2 appeared to have significantly less pain than group 1 at baseline, representing both NGRS pain scores at the start of the cycles (Z(770) = −2.41; *p* = 0.016) and end of the cycles (Z(614.3) = −2.57; *p* = 0.010). However, these were not significant when examined using adjusted *p*-values and so may have occurred by chance.

The NGRS pain scores for group 1 decreased significantly over the course of the study both before (Z(1158) = −5.62; adjusted *p* < 0.001) and after (Z(1067) = −3.57; adjusted *p* = 0.009) BONT-A injection. For group 1, the model demonstrates a mean difference in NGRS score of −0.65 at the start of each cycle and of −0.49 at the end of each cycle after controlling for covariates (see Figure A2 in the Appendix B).

The interactions between group and cycle number indicate that group 2 had a significantly slower decrease (rate of change in slope) in the start-of-cycle pain scores than group 1 (Z(1140) = 3.98; adjusted *p* < 0.001), with a 0.5 difference at each cycle. This was also true for the end-of-cycle pain scores (Z(1049) = 3.08; adjusted *p* = 0.049), with a difference of 0.45 at each cycle. The slope for group 3 was not significantly different from group 1 for both the start-of-cycle or end-of-cycle visits (adjusted *p* = 0.140 and 0.165, respectively).

These differences are quantified in Figure A2, which shows the plot of the interactions between group and cycle number for the three groups. The tables show the fitted line for each of the three groups, as derived from the slope of the plot, after taking into account the patient-related covariates in the mixed-effects model. The fitted line can be compared to the raw means in Table A4.

For the start-of-cycle visits (before injection), group 1 had an average decrease in NGRS pain score of 1.3 between cycle 1 and cycle 3, and group 3 had an average reduction of 2.2 by the end of cycle 8, and group 2 had only an average reduction of 0.6 between cycle 1 and cycle 5.

For the end-of-cycle visits (after injection), group 1 had an average decrease in pain score of 1.0 by the end of cycle 3, and group 3 had an average reduction of 0.8 by the end of cycle 8, while group 2 had an average decrease of only 0.2 by the end of cycle 5.

### 2.6. Predictors of Cycle Frequency

Finally, turning the question the other way around, we examined the factors that may be associated with either a requirement for fewer than 4 cycles or more frequent injections (i.e., 6–8 cycles) during the 2-year period.

Table 4 shows the logistic regression analyses of predictors of cycle-frequency group membership between groups 1 and 3. The two groups were not different in terms of age, gender, severity of pain at baseline, distribution or chronicity of spasticity (adjusted *p* > 0.713). However, patients who had received BoNT-A injection(s) for upper limb spasticity before the study had significantly higher odds of having more frequent injections (6–8 cycles) compared to those who were naïve to BoNT-A (Z = 4.34, adjusted *p* < 0.001) with an odds ratio of 5.7, with a 95% CI [2.7, 13.2].

## 3. Discussion

This post hoc analysis of the ULIS-III study evaluated the longitudinal effects of repeated injections of BoNT-A for upper limb spasticity-related pain in the context of routine clinical practice over a 2-year period. It demonstrated that spasticity-related pain is a priority goal for repeated treatments with BoNT-A in approximately two-thirds of patients, and that they continue to show a significant response in terms of both goal attainment and pain reduction (i.e., change in score) following each injection up to at least seven injection cycles.

### 3.1. Response to Repeated Injections

Although the magnitude of response with each injection tended to diminish slightly over successive cycles and repeated-measures ANOVAs were borderline significant, the effect sizes between cycles were small (0.01–0.13; see Table 2c) and therefore unlikely to be clinically significant. This means that BoNT-A continued to produce a reduction in pain scores ranging from 1.8 to 2.7 after each injection of BoNT-A (see Table A2), and the response did not diminish significantly at a clinical level even after eight injections.

### 3.2. Serial Changes in Pain Severity

An initial exploration of the serial changes in pain score showed that patients who had more frequent injections tended to have proportionately higher pain scores at baseline. In addition, there was a clear trend towards a progressive reduction in pain scores both at the start and end of each cycle, suggesting some cumulative effect of repeated injection on pain over 2 years. Throughout the study, patients had an average reduction in pain score between 0.6 and 2.2 for start-of-cycle scores (see Figure A2), which indicates that repeated administration of BONT-A for upper limb spasticity is associated with a significant progressive decrease in pain over the 2-year period after accounting for other variables. This is a moderate effect size which may therefore be significant at a clinical level.

### 3.3. Patient-Related Predictors of Pain Reduction and the Frequency of Injection

Mixed-effects models were used to explore patient-related factors that may predict pain scores before and after injection and the rate of change across successive cycles. When examined in three groups of cycle frequency during the 2-year period, group 1 (who had a maximum of 3 injections) had a sharper decrease in pain scores across successive cycles compared to patients in groups 2 (maximum of 4–5 injections) and 3 (maximum of 6–8 injections), who had a slower decrease across cycles. However, when other patient-related factors were taken into account, this difference was only statistically significant for group 2.

Taken together, the above findings suggest that those who required more frequent injections may have had more resistant pain, but nevertheless, they still continued to show significant pain response to each BoNT-A even after seven injections. Using logistic regression, we explored possible factors that might predict which patients required more frequent injections (6–8 cycles) in comparison to those who required only 1–3. Age, gender, time since onset, severity of spasticity and DAS pain score at baseline did not significantly impact injection frequency, but those requiring 6–8 injections were significantly more likely to have previous injections with BoNT-A for upper limb spasticity prior to the start of the study, suggesting that more of them were already on established programmes of treatment.

### 3.4. Comparison of Findings with Other Studies

Although other studies have demonstrated that pain is a common goal for treatment of upper limb spasticity involving single cycles of BoNT-A [41,42], and several authors have explored the benefits of repeated injection on outcomes such as tone, range of movement, function and quality of life [45,46,47], few have explored the effects on spasticity-related pain over repeated injection cycles.

Wissel et al., 2021 [27] reported a pooled analysis of six studies, which also demonstrated progressive improvement in pain over repeated cycles, but they used the DAS, rather than a rating on a scale of 0–10, so their findings are not directly comparable to this study. Findings from the ASPIRE study of onabotulinumum toxin for upper limb spasticity [48] reported average changes in pain score ranging from −0.5 to −1.2 over eight cycles, which are somewhat smaller than the changes in the descriptive statistics reported here (which range from −1.8 to −2.8 across the cycles (See Table A2)). Riberto et al., 2022 [31] recorded improvements in hemiplegic shoulder pain on a scale of 0–10 following just two cycles of BoNT-A, demonstrating a mean change of −2.9 from baseline to the end of cycle 1 and −3.5 from baseline to the end of cycle 2. These are comparable with our descriptive findings of a reduction in pain score of −2.8 from baseline to the end of cycle 1 and −3.2 to the end of cycle 2 in the first two cycles. However, to our knowledge, ours is the first study to examine changes in pain score within individuals or to explore the presentation characteristics that may impact or predict the response to pain over repeated cycles.

### 3.5. Important Messages for Clinicians Managing Upper Limb Spascticity

This analysis used a triangulation of different techniques, including visual inspection, descriptive statistics and a variety of statistical analyses, to derive a number of important messages for clinicians who manage upper limb spasticity and those who commission and plan services:Reduction in pain is a common goal for treatment in patients with upper limb spasticity and BoNT-A continues to provide relief from spasticity-related pain over repeated injections.There is evidence for both a short-term response to each injection and a longer-term cumulative effect of reduced pain over successive cycles, both of which are likely to be clinically significant.Some patients appear to require more frequent injections to manage their symptoms. Those who required more frequent injections had more severe and resistant pain over the course of the study and were more likely to have had previous treatment with BoNT-A, but as of yet, there is no clear algorithm for determining injection frequency based on patient characteristics or presentation. This can only be determined empirically over time.Patients who required only 1–3 injections tended to have a more rapid pain relief over successive cycles, and it is possible that some of these made such significant improvement that no further injections were required, although we cannot be certain about this from the data.The study was limited to 2 years for pragmatic reasons, but this does not mean that treatment should be limited to that period. Patients who require repeated injections to manage spasticity-related pain may require life-long treatment.

Determination of a minimally important clinical difference (MCID) is challenging in the context of pain, as it is highly dependent on individual circumstances and perceptions, and there are no definitive data on an MCID for the NGRS score in the context of upper limb spasticity. However, the advantage of measuring pain in parallel with pain goals, and in the context of repeated cycles, is that patients literally become experts in what they can expect to achieve from repeated injections and in setting the target goals for each treatment. The ULIS-III study has provided a rich source of data with which to explore, in detail, the relationship between pain reduction and SMART statements for pain-related goals. This will be the subject of future analyses.

### 3.6. Strengths and Weaknesses

The authors acknowledge a number of strengths and weaknesses of this study.

Strengths include the large size of the study, conducted in real-life clinical practice with wide international representation, including all aetiologies and all BoNT-A products, which helps to ensure the generalisability of the findings.Limitations in the study’s design include the lack of a control group and the fact that some countries only had a few active sites, while others had several, so the findings may not be truly representative. Additional biases may include factors such as clinician expertise on treatment decisions, individual injector beliefs or habits or external prescribing restrictions that limit the number of permitted injections.The ULIS-III study was conducted in real-life clinical practice and spasticity-related pain may be multifactorial. Within the dataset, there is no specific information defining the specific cause(s) of pain—merely the clinical observation that pain management was a personal goal for treatment within that cycle.The ULIS-III study has provided a large and rich dataset for post hoc analyses. The wider the net is cast in a post hoc analysis, the more chance there is of introducing statistical error through multiple tests. In this study, we have explored in some detail the impact of patient characteristics (such as gender, age, severity distribution and chronicity of spasticity) on pain outcomes from repeated injection. However, we have not yet explored other possible variables, for example, related to treatment approach—such as the dose, agent, injection technique (e.g., number and distribution of muscles, use of targeting techniques, etc.) or concomitant therapies—all of which would need to be addressed in future analyses.

## 4. Conclusions

Despite the acknowledged limitations, findings from this study demonstrate that a reduction in spasticity-related pain is a common goal of treatment with BoNT-A for upper limb spasticity, and that repeated injections over a 2-year period continue to produce a significant reduction in spasticity-related pain, regardless of factors such as age, previous injection history, time since onset, severity and distribution of spasticity.

## 5. Materials and Methods

### 5.1. Study Design and Participants

This is a retrospective secondary analysis of data collected from the ULIS-III study (NCT02454803). Full details of the methodology have been previously described [40,42].

In brief, ULIS-III was an international observational, prospective, longitudinal cohort study following patients with upper limb spasticity over 2 years treated through integrated programmes incorporating BoNT-A injections and physical management, as delivered in real-life clinical practice. It was conducted in 58 centres across 14 countries and four continents.

Specialist centres recruited up to 30 consecutive adult patients (≥18 years old) in whom a decision had been made to treat upper limb spasticity with BoNT-A in the course routine clinical practice. Patients were managed in accordance with local marketing authorisation and as per normal practice for each centre using any licenced BoNT-A formulation.

Each patient was followed for a period of 2 years, during which they may have had more than one cycle of BoNT-A injection. Treatment was goal-directed, and patients could change their primary and/or secondary goals at the baseline visit of each cycle. Ethical approval and written informed consent to the recording of anonymous data were obtained in countries where this was required.

### 5.2. Outcome Assessment and Measures

Patients were assessed at baseline and at each follow-up visit throughout the 2-year follow-up period using the Upper Limb Spasticity Index (ULSI). This records GAS alongside a limited set of standardised measures, selected specifically according to the priority goal area(s) for treatment in each cycle.

Goals for treatment were captured using the Goal Attainment Scaling Evaluation of Outcome for Upper Limb Spasticity (GAS-eous) tool [40], which uses a structured approach to goal-setting and evaluation of goal attainment, which is assimilated to yield a GAS T-score utilising the GAS light method [50]. Patients could set up to 3 goals (1 primary and 2 secondary) per cycle. The GAS T formula is designed such that if goals are achieved as expected, the mean GAS T-score will be 50 with a standard deviation of +/−10 [34]. In the context of spasticity management, the minimal clinically important change in GAS T-score from baseline is reported to be 10 [37].

Of the 953 patients included in the ‘effectiveness population’, for the purpose of this secondary analysis, we were interested in those patients who identified pain as a goal for treatment at some point during the study period. The standardised measure for pain was the NGRS, measured on a range from 0 (‘no pain’) to 10 (‘pain as bad as can be’) [40].

### 5.3. Statistical Analysis

Patients were included in the ‘pain subgroup’ if they had at least one pain goal (primary or secondary) in relation to spasticity-related pain for at least one cycle. Each cycle is defined as the before and after assessment of each injection of BONT-A, and therefore, there are two visits per cycle. The response to each injection in terms of pain reduction is reflected in the ‘change score’ (i.e., the difference between the start-of-cycle and end-of-cycle NGRS pain scores).

Because ULIS-III was an observational cohort study conducted in real-life clinical practice (rather than as part of a structured trial with pre-set data points), patients who had an initial pain-related goal did not necessarily have pain-related goals (and therefore pain measures) in successive or subsequent cycles, which poses a challenge for longitudinal analysis. The first step of the evaluation was therefore an exploratory descriptive and visual analysis to examine trends in the data, which was carried out before applying more sophisticated statistical analysis techniques.

Previous analyses have demonstrated that patients received between 1 and 8 cycles during the study. We wished to explore any differences between those who required ‘more frequent’ versus ‘less frequent’ injections for pain-related goals. Patients were therefore analysed in three groups based on the maximum number of injections received (cycle frequency) during the 2-year period: group 1 had 1–3 cycles only, group 2 had 4–5 cycles and group 3 had 6–8 cycles.

No outliers were removed in this study nor were any transformations made to the original variables. Missing data were not imputed and *p*-values in this study were adjusted to control for multiple comparisons within each set of analyses, using the Holm–Bonferroni method [51]. (All *p*-values given in the text are adjusted.)

Analysis of the data was completed in R (version 4.3.2). The primary aim of this study was to identify how GAS T-scores and NGRS pain scores change over the course of repeated cycles for those patients whose goals for treatment included a reduction in spasticity-related pain. Mean before and after scores, as well as the change scores, for each cycle were calculated with 95% bootstrapped confidence intervals (CIs), each with 1000 samples, and were compared using paired *t*-tests and stratified by cycle-frequency group. A repeated-measures ANOVA was used to examine the effect size by number of cycles with Greenhouse–Geisser correction for non-sphericity [52].

A mixed-effects model was used to examine variables that predicted NGRS pain scores (at the start and end of each cycle, separately) while controlling for their confounding effect. We conducted two analyses, one for the start-of-cycle visits (before each BoNT-A injection) and one for the end-of-cycle visits (after injection). The observations represented participants and cycle number, as opposed to participants alone. Because this study was an AB design, with a maximum of eight cycles, the observations in the data do not fulfil the independence assumption. We controlled for dependency between observations over time by including a random-effect term for each individual. Using a model that analysed by cycle and person provided more power in detecting significant differences between the predictor variables and response variables in this study. We also added an interaction term to both models to account for different rates of change in the NGRS pain score for each group across the cycles.

Based on experience from previous analyses [49], we hypothesised that patient-related factors which may predict (or confound) pain reduction over the course of the study might include age, gender, chronicity of spasticity (years since onset), previous treatment with BoNT-A, severity and distribution of spasticity (as measured by the composite MAS for the proximal (elbow and shoulder) [42] and distal (hand, wrist and finger) muscles) and the number of cycles (or visits) required during the 2-year period of the study.

We applied a logistic regression analysis to explore any patient-related factors that may predict a requirement for either a small (<4 cycles) or large (6–8) number of cycles during the 2-year period. We once again hypothesised that these may include age, gender, chronicity of spasticity, previous treatment with BoNT-A, severity and distribution of spasticity and the severity of pain at baseline (as measured by the pain item of the DAS at baseline).

## Figures and Tables

**Figure 1 toxins-17-00117-f001:**
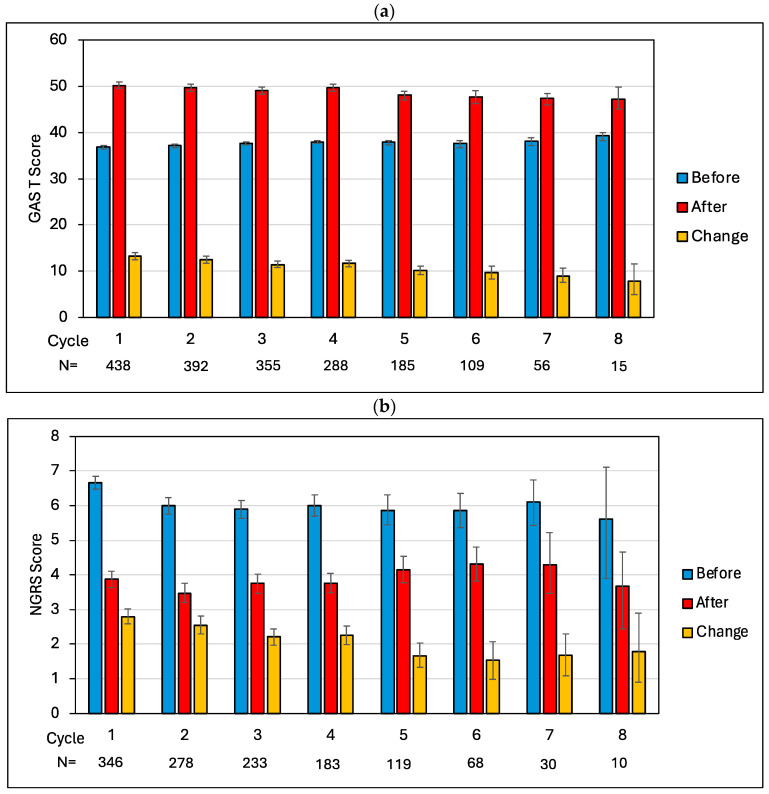
(**a**) GAS T-scores for overall goal attainment across each cycle, and (**b**) NGRS scores for pain severity. Note: GAS = Goal Attainment Scaling; NGRS = Numbered Graphic Rating Scale (range 0–10).

**Figure 2 toxins-17-00117-f002:**
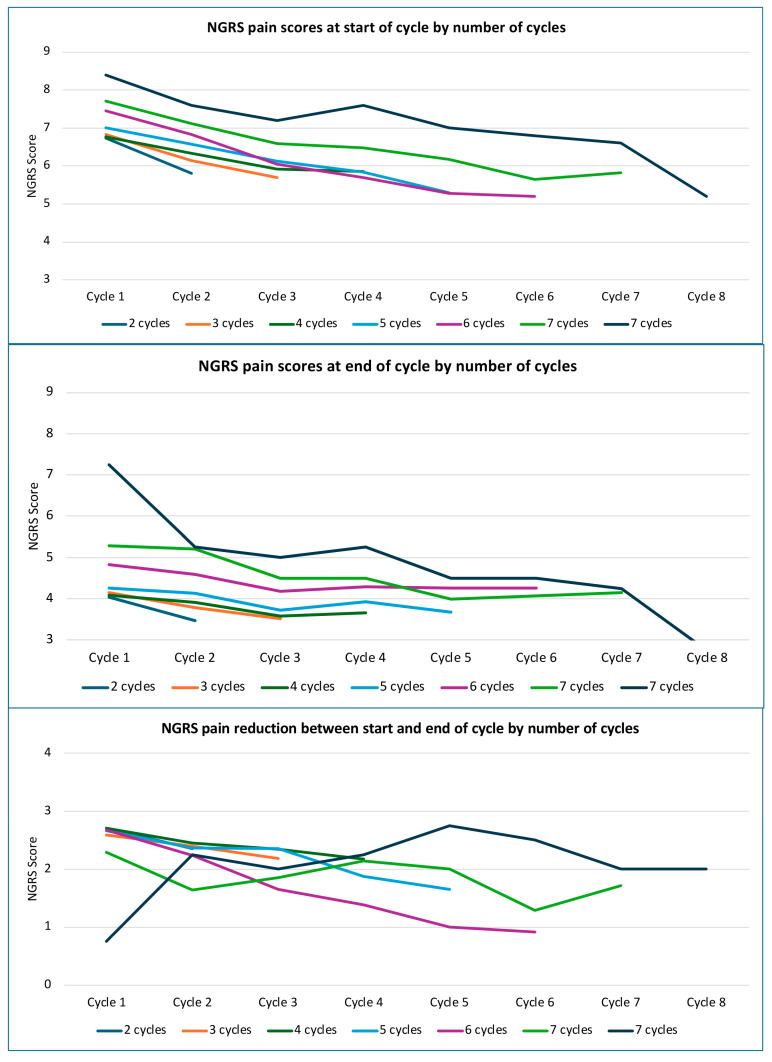
Mean NGRS pain scores with the number of cycles.

**Figure 3 toxins-17-00117-f003:**
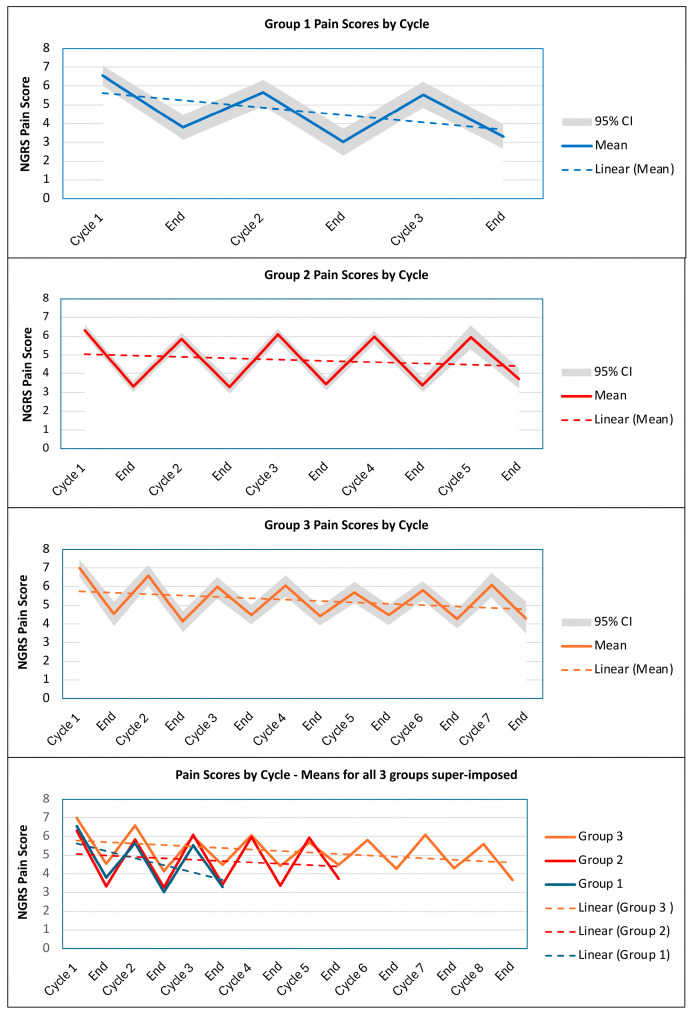
Line graphs of the mean (bootstrapped 95% CI) pain scores at the start and end of each cycle within the three ‘cycle-frequency’ groups.

**Table 1 toxins-17-00117-t001:** Demographics of the overall effective population and the pain subgroup.

	Effective Population (N = 953)		Pain Subgroup (N = 438)		Adjusted *p*-Value *
Demographic	N	%	N	%	χ^2^
**Sex—n, %**					0.109
Male	537	56.3%	226	51.6%	
Female	416	43.7%	212	48.4%	
**Age—n, %**					1.0
<51 years	364	38.2%	158	36.1%	
51–60 years	254	26.7%	124	28.3%	
>60 years	335	35.2%	156	35.6%	
**Type of injury—n, %**					0.704
Acquired brain injury	870	91.3%	391	89.3%	
Spinal cord injury	15	1.6%	6	1.4%	
Congenital	44	4.6%	24	5.5%	
Progressive neurological	20	2.1%	13	3.0%	
Other	4	0.4%	4	0.9%	
**Aetiology of acquired brain injury—n, %**					1.0
Vascular (stroke)	786	90.3%	364	93.1%	
Trauma	71	8.2%	25	6.4%	
Hypoxic	25	2.9%	11	2.8%	
Tumour	19	2.2%	8	2.0%	
Inflammatory/infective/other	52	6.0%	30	7.7%	
**Time since onset of injury**					1.0
<1 year—n, %	145	15.2%	74	16.9%	
>1 year—n, %	769	80.7%	344	78.5%	
Years—mean (SD)	7.6	(9.4)	7.6	(9.6)	
Missing—n, %	39	4.1%	20	4.6%	
**Previous injections with BoNT-A for upper limb spasticity—n, %**					1.0
No (Naïve)	318	33.4%	151	34.5%	
Yes (Non-Naïve)	635	66.6%	287	65.5%	
**Distribution of spasticity—n, %**					**<0.001**
Focal	190	19.9%	60	13.7%	
Regional	763	80.1%	378	86.3%	
**Baseline pain and spasticity**	**Mean**	**95% CI**	**Mean**	**95% CI**	** *t* ** **-test**
Baseline DAS pain score	1.1	1.0, 1.1	1.8	1.7, 1.9	**<0.001** **
Proximal composite MAS score	3.7	3.6, 3.8	3.9	3.7, 4.0	**0.014**
Distal composite MAS score	6.1	5.9, 6.2	6.1	5.9, 6.3	1.0

DAS = Disability Assessment Scale; MAS = Modified Ashworth Scale; CI = bootstrapped confidence interval. χ^2^ = chi square. * Adjusted *p*-values for the difference between the subgroups of the effective population with one or more pain-related goals (n = 438) and with no pain-related goals (n = 497). ** Levene’s test was statistically significant.

**Table 2 toxins-17-00117-t002:** Repeated-measures ANOVAs for the start of the cycle pain scores, end of the cycle pain scores and the change in pain scores by number of cycles.

(a) NGRS Scores at the Start of Cycle						
Cycles	Effect Size (η^2^)	df1	df2	F	*p*-Value	Adjusted *p* *
2	0.17	1	230	46.6	**<0.001**	**<0.001**
3	0.14	2.0	320.6	25.4	**<0.001**	**<0.001**
4	0.08	2.6	299.5	9.6	**<0.001**	**<0.001**
5	0.16	3.1	227.6	13.9	**<0.0001**	**<0.001**
6	0.27	3.7	142.9	14.3	**<0.001**	**<0.001**
7	0.18	3.7	59.3	3.4	0.016	0.160
8	0.31	7	28	1.8	0.129	0.742
**(b) NGRS Scores at the End of Cycle**						
**Cycles**	**Effect Size (η^2^)**	**df1**	**df2**	**F**	** *p* ** **-Value**	**Adjusted *p***
2	0.08	1	222	19.5	**<0.001**	**<0.001**
3	0.06	2.0	298.5	9.4	**<0.001**	**0.002**
4	0.03	2.7	293.0	3.8	0.014	0.154
5	0.04	3.3	227.7	2.8	0.036	0.324
6	0.04	4.0	132.1	1.2	0.297	0.891
7	0.11	3.1	40.8	1.6	0.192	0.768
8	0.65	7	21	5.6	**0.001**	**0.013**
**(c) Change in NGRS Scores in Each Cycle**						
**Cycles**	**Effect Size (η^2^)**	**df1**	**df2**	**F**	** *p* ** **-Value**	**Adjusted *p***
2	0.01	1	221	4.0	0.047	0.376
3	0.01	2	294.8	2.2	0.114	0.742
4	0.02	2.7	288.7	2.1	0.106	0.742
5	0.06	3.2	226.3	4.5	0.004	0.048
6	0.14	3.5	116.9	5.5	**0.001**	**0.011**
7	0.03	3.0	39.5	0.4	0.732	1
8	0.13	7	21	0.4	0.862	1

Note: η^2^ = generalised eta squared; df = degree of freedom between groups (df1 for number of cycles—1; df2 sample size—number of groups). F = F test statistic which identifies how unusual the differences are, given the null hypothesis that all means are equal. Decimals are used to correct for a violation of assumptions. Adjusted *p* = *p*-value adjusted for the number of tests in this analysis using the Holm–Bonferroni method.

**Table 3 toxins-17-00117-t003:** Mixed-effects model of NGRS pain score, with the individual as a random effect.

(a) Start of Cycle						
Variable	B	SE	df	Z	*p*-Value	Adjusted *p* *
**Intercept**	6.45	0.33	521.6	19.75	**<0.001**	**<0.001**
**Group** Group 2	−0.55	0.23	770.0	−2.41	**0.016**	0.312
Group 3	−0.09	0.27	668.9	−0.34	0.738	1
**Age**						
51–60 years	0.12	0.21	418.2	0.58	0.560	1
61+ years	−0.08	0.20	431.5	−0.41	0.679	1
**Sex**						
Female	0.32	0.17	424.9	1.94	0.053	0.953
**Time since injury**	<0.01	<0.01	439.5	−0.15	0.879	1
**Previous BoNT-A for ULS**						
Yes	0.09	0.19	456.3	0.50	0.618	1
**Distribution of spasticity**						
Regional	0.09	0.24	442.6	0.38	0.704	1
**Severity of spasticity** MAS proximal	<0.01	<0.01	433.4	1.30	0.193	1
MAS distal	<0.01	<0.01	409.4	0.62	0.538	1
**Cycle number**	−0.65	0.12	1158	−5.62	**<0.001**	**<0.001**
Group 2*cycle number	0.50	0.13	1140	3.98	**<0.001**	**0.002**
Group 3*cycle number	0.33	0.12	1142	2.73	**0.006**	0.140
**(b) End of cycle**						
**Variable**	**B**	**SE**	**df**	**Z**	** *p* ** **-Value**	**Adjusted *p* ***
**Intercept**	3.40	0.36	467.6	9.34	**<0.001**	**<0.001**
**Group**						
Group 2	−0.63	0.25	641.3	−2.57	**0.010**	0.206
Group 3	−0.31	0.29	558.4	−1.06	0.290	1
**Age**						
51–60 years	0.11	0.24	394.1	0.48	0.632	1
61+ years	0.08	0.23	405.3	0.35	0.729	1
**Sex**						
Female	0.34	0.19	398.8	1.79	0.074	1
**Time since injury**	<0.01	<0.01	407.4	−0.23	0.821	1
**Previous BoNT-A for ULS**						
Yes	−0.09	0.21	420.3	−0.46	0.647	1
**Distribution of spasticity**						
Regional	0.46	0.27	411.4	1.69	0.092	1
**Severity of spasticity** MAS proximal	<0.01	<0.01	403.8	0.72	0.472	1
MAS distal	<0.01	<0.01	388.1	0.59	0.553	1
**Cycle number**	−0.49	0.14	1067	−3.57	**<0.001**	**0.009**
Group 2*cycle number	0.45	0.15	1049	3.08	**0.002**	0.049
Group 3*cycle number	0.38	0.14	1053	2.66	**0.008**	0.165

Note: B = regression coefficient; SE = standard error; df = degree of freedom; Z = Z score; *p* * = adjusted *p*-value for this analysis; BoNT-A = botulinum toxin A; MAS = Modified Ashworth Scale.

**Table 4 toxins-17-00117-t004:** Logistic regression of predictors of group membership for group 1 and group 3.

Variable	B	SE	Z	*p*-Value	Adjusted *p* *
**Intercept**	−1.43	0.79	−1.80	0.071	0.713
**Age**					
51–60 years	−0.02	0.42	−0.05	0.965	1
61+ years	−0.15	0.39	−0.39	0.697	1
**Sex**					
Female	−0.24	0.32	−0.76	0.447	1
**DAS pain score at baseline**					
1	0.46	0.66	0.70	0.487	1
2	0.24	0.58	0.41	0.679	1
3	0.05	0.62	0.09	0.931	1
**Previous BoNT-A for ULS**					
Yes	1.74	0.40	4.34	<0.001	<**0.001**
**Severity of spasticity** MAS proximal	−0.02	0.11	−0.23	0.817	1
MAS distal	−0.07	0.07	−1.01	0.312	1
**Years since stroke**	−0.01	0.02	−0.34	0.736	1

Note: B = regression coefficient; SE = standard error; df = degree of freedom; Z = Z score; * adjusted *p*-value for this analysis; MAS = Modified Ashworth Scale; DAS = Disability Assessment Scale (1 = mild, 2 = moderate, 3 = severe disability); ULS = upper limb spasticity.

## Data Availability

Data sharing is restricted due to the limits of patient confidentiality and consent. Further details on Ipsen’s sharing criteria, eligible studies and the process for sharing are available here: https://vivli.org/ourmember/ipsen/ (accessed on 23 February 2025). Any requests should be submitted to www.vivli.org for assessment by an independent scientific review board.

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
