# Peer review of "Pain Reduction with Repeated Injections of Botulinum Toxin A in Upper Limb Spasticity: A Longitudinal Analysis from the ULIS-III Study"

_toxins, 2025, doi:10.3390/toxins17030117_

Round 1
Reviewer 1 Report
Comments and Suggestions for Authors
The study examines whether pain goals are sustained over repeated injection cycles through a retrospective secondary analysis. It reports serial changes in pain severity and explores predictors of pain reduction and injection frequency. While the project has the potential to contribute valuable knowledge to the field, supported by a large sample size, the results appear limited in terms of their clinical importance. Substantial revisions are necessary to address several concerns.
Title
The title should be refined to reference upper-limb spasticity and pain for consistency.
Abstract
The abstract’s readability can be improved. The sentence, “The ULIS-III study was a prospective international, observational, longitudinal study conducted in real-life clinical practice over a 2-year period,” contrasts with, “In this retrospective secondary analysis, we examine whether pain goals continued to be met over repeated injection cycles. We report serial changes in pain severity and explore predictors of pain reduction and injection frequency.” This discrepancy should be resolved. Additionally, the results section lacks p-values or references to significant effects, which makes it unsuitable. The conclusion must explicitly refer to both upper limb pain and spasticity.
Key Contributions
This section should focus on the primary contributions, specifically referencing upper-limb pain and spasticity.
Introduction
The introduction should clearly outline the study’s contribution to the existing literature, especially about the findings of the primary ULIS-III study. Briefly explain the rationale for using Botulinum toxin A for pain reduction. Conclude the introduction by stating the study’s hypothesis and the relevance of performing this secondary analysis.
Results
In the demographics section, phrases like “broadly similar for the Pain subgroup and the Effectiveness population from which it was drawn” are too vague. Statistical comparisons should be reported.
The results section requires extensive revision to improve clarity, conciseness, and readability. There are numerous instances where findings are described in a non-statistical and confusing manner. Avoid descriptive approaches such as, “Pain scores for Groups 1 and 2 are broadly similar at baseline,” or, “There also appears to be a different intercept for Group 2 compared with Groups 1 and 3 in terms of the fitted line.” Instead, focus on reporting statistically significant results with appropriate tests.
Statements like, “The results suggest people had, on average, an NGRS score of 6.45 at the start of the first cycle, reducing to 3.4 at the end of that cycle, indicating an average reduction of just over 3 points after the first injection of Botulinum toxin, after taking the patient-related factors into account,” do not belong in the results section. Such details should be reserved for the discussion.
Discussion
The discussion must consistently refer to upper limb spasticity and pain. Focus on significant statistical results and structure the discussion using subheadings aligned with the results and the study aims stated in the introduction. Address biases, such as the influence of clinician expertise on treatment decisions, which may vary significantly across centres.
Methods
Clearly define the criteria used to attribute pain to spasticity. This is a critical concern, as multiple centres were involved, and consistency must be ensured.
Some methodological details include, “In brief, ULIS-III was an international observational, prospective, longitudinal cohort study following patients with upper-limb spasticity over 2 years treated through integrated programs incorporating BoNT-A injections and physical management as delivered in real-life clinical practice. It was conducted in 14 countries across four continents,” belongs in the introduction when referring to the main study.
Clarify whether the 2-year extension was included in the ethical committee and clinical trial approvals, and provide the number of ethical approvals.
Comments on the Quality of English LanguageThe quality of the English Language must be improved in readability and conciseness.
Author Response
Thank you very much for your helpful comments, which we have done our best to address
Please see the attached file

Reviewer 2 Report
Comments and Suggestions for Authors
Dear authors
The paper is fine, but it could be better. Kindly follow my comments and do changes in the document as you could.
The Title
1- The title is long, kindly use a shorter and attractive title.
The Abstract
2- (N=438) line 13 is not clear what it means and one need to go down on the text to understand what you mean.
3- In a sentence like “Pain-goals were common and achieved in 65-75%” is not clear what you mean with bain-goals and what is the numbers (as %) mean or related to what! Again, one need to go down to get more idea. Kindly by writing a paper you need to consider readers from other related domains whom might not familiar with a term like “Pain-goals”
Keywords
4- Kindly use the keywords “goal attainment scaling” in more frequency in the entire of the text.
Introduction
5- Kindly add one paragraph to explain the pain mechanism in your case study. Kindly, include some prospective as general information as well as some other for the specialist and who could the injection of the botulinum toxin A might interfere or do other mechanism(s) to reduce the pain(s).
6- You might also need to highlight the differences between the pain and the conditions that lead to its appearance. And in which side stand the botulinum toxin A in one of them or in both sides.
Results
7- In line 67 “had previously had BoNT-A injections for ULS” you might replace the second “had” with obtained or another suitable verb.
8- In line 134 kindly avoid use word like very.
9- You need to add Y axis to Figure 2.
Discussion
10- In line 327 you need to change the word ‘’ People’’ with patients.
Materials and Methods
11- Line 395, 396 kindly if you use number, so kindly include the unit, persons, % etc. Kindly, do that in the entire text.
12- You need to describe the name of the software you have used and its version.
References
13- You need to add at least 7 more references
Additional improvement
14- I have observed that you did not use the abbreviation style correctly. You repeat the use the words and their abbreviations in the same time. Kindly correct that in the entire text. For example, you have repeated (numbered graphic rating scale (NGRS)) four times.
15- Kindly, don’t use the overestimated words like very, many, etc.
with my pleasure

The English is fine. But Kindly, don’t use the overestimated words like very, many, etc.
Author Response

(The authors gave the same response as above.)
